# Equine alveolar macrophages and monocyte-derived macrophages respond differently to an inflammatory stimulus

Heng Kang[1], Gary Kwok Cheong Lee[1,2], Dorothee Bienzle[1], Luis G. Arroyo[3], William Sears[4], Brandon N. Lillie[1], Janet Beeler-Marfisi[1]*

1 Department of Pathobiology, University of Guelph, Guelph, Ontario, Canada, 2 IDEXX Laboratories Pty. Ltd., Rydalmere, New South Wales, Australia, 3 Department of Clinical Studies, University of Guelph, Guelph, Ontario, Canada, 4 Department of Population Medicine, University of Guelph, Guelph, Ontario, Canada

* jbeelerm@uoguelph.ca

**Data Availability Statement:** All relevant data are within the paper and its Supporting Information files.

## Abstract

Alveolar macrophages (AMs) are the predominant innate immune cell in the distal respiratory tract. During inflammatory responses, AMs may be supplemented by blood monocytes, which differentiate into monocyte-derived macrophages (MDMs). Macrophages play important roles in a variety of common equine lower airway diseases, including severe equine asthma (SEA). In an experimental model, an inhaled mixture of *Aspergillus fumigatus* spores, lipopolysaccharide, and silica microspheres (FLS), induced SEA exacerbation in susceptible horses. However, whether equine AMs and MDMs have differing immunophenotypes and cytokine responses to FLS stimulation is unknown. To address these questions, alveolar macrophages/monocytes (AMMs) were isolated from bronchoalveolar lavage fluid and MDMs derived from blood of six healthy horses. Separately, AMMs and MDMs were cultured with and without FLS for six hours after which cell surface marker expression and cytokine production were analyzed by flow cytometry and a bead-based multiplex assay, respectively. Results showed that regardless of exposure conditions, AMMs had significantly higher surface expression of CD163 and CD206 than MDMs. Incubation with FLS induced secretion of IL-1β, IL-8, TNF-α and IFN-γ in AMMs, and IL-8, IL-10 and TNF-α in MDMs. These results suggest that AMMs have a greater proinflammatory response to in vitro FLS stimulation than MDMs, inferring differing roles in equine lung inflammation. Variability in recruitment and function of monocyte-macrophage populations warrant more detailed in vivo investigation in both homeostatic and diseased states.

## Introduction

Alveolar macrophages (AMs) are the most numerous innate immune cell in the healthy distal lung, and reside on the luminal surface of the alveolus [1, 2]. These macrophages arise during embryogenesis, populate the alveoli after birth, and self-renew throughout the life of the animal [3–5]. In the steady state, AMs perform homeostatic functions including surfactant

**Funding:** JBM was funded by Equine Guelph Grant #EG 2017 01, and DB was funded by Ontario Ministry of Agriculture, Food and Rural Affairs Grant #27364. Equine Guelph website: https://www.equineguelph.ca/#gsc.tab=0 Ontario Ministry of Agriculture, Food and Rural Affairs website http://omafra.gov.on.ca/english/ The funders had no role in study design, data collection and analysis, decision to publish, or preparation of the manuscript.

**Competing interests:** The authors have declared that no competing interests exist.

clearance and immune modulation to enable optimal gas exchange [2, 6]. Additionally, AMs rapidly recognize respirable particulates and pathogens, and initiate inflammatory responses [6, 7]. In the barn environment, respirable materials capable of inciting an AM response include fungal spores, lipopolysaccharide (LPS) from environmental gram-negative bacteria, and soil-origin dusts [8–10]. Indeed, an inhalational challenge model composed of fungal spores, LPS and silica microspheres (FLS), induced exacerbation of severe equine asthma (SEA) in susceptible horses–identifying these components as critical in inducing an inflammatory response [8].

Surface expression of CD163, the hemoglobin-haptoglobin scavenger receptor, and CD206, the mannose receptor, is characteristic of equine AMs as they naturally express both receptors but can alter this expression pattern in response to inhaled particulates [11]. Although increased AM expression of CD163 and CD206 has been associated with an anti-inflammatory phenotype in conjunction with severe, but non-neutrophilic, asthma in humans [12–18], it is uncertain whether the same is true of equine AMs in the context SEA. In contrast, blood monocytes arise from bone marrow precursors and infiltrate tissues, including the lung [3, 19, 20]. Particularly during an inflammatory response, monocytes differentiate into either pro- or anti-inflammatory monocyte-derived macrophages (MDMs) in response to local signals [21, 22]. Three subpopulations of monocytes have been identified: so-called classical monocytes ($CD14^{++}CD16^-$), intermediate monocytes ($CD14^{++}CD16^+$), and non-classical monocytes ($CD14^+CD16^{++}$) [23]. In horses, the majority of monocytes in blood were deemed classical and in bronchoalveolar lavage fluid (BALF), intermediate [24, 25], and a decrease in CD14 + macrophage proportions was detected in SEA-affected horses [24].

Macrophages initiate an inflammatory cascade by recognizing pathogen-associated molecular patterns and damage-associated molecular patterns via pattern recognition receptors [26]. For example, macrophages express dectin-1, a member of the C-type lectin-like receptor family, which binds β-glucan on swelling fungal spores and triggers a proinflammatory response to these potential pathogens [27, 28]. Binding of LPS by TLR-4, and its co-receptor myeloid differentiation factor 2, results in activation of transcription factors such as NF-κB, leading to the production of both pro- and anti-inflammatory cytokines including tumor necrosis factor alpha (TNF-α), interleukin (IL)-1β, IL-8, and IL-10 [9, 29, 30]. This production of opposing cytokines highlights the dual role of macrophages in inciting and resolving an inflammatory response to maintain homeostasis [31].

Under homeostatic conditions, the majority of monocytes/macrophages in alveoli are mature AMs, and the minority are intermediate monocytes and monocyte-lymphocyte cells [25, 32]. However, this ratio can be altered dramatically in disease conditions. In humans, increased airway classical monocytes were a feature of neutrophilic asthma [33], and in a mouse model of LPS-induced pulmonary inflammation, there was wide-spread replacement of AMs by recruited monocytes–the latter amplified the inflammatory response by producing abundant cytokines and chemokines [34]. Although AMs and MDMs both figure in the pathogenesis of a variety of pulmonary inflammatory diseases including asthma, acute lung injury, acute respiratory distress syndrome, and fibrosis [16], only rarely has it been considered that tissue-resident macrophages and MDMs may have differing responses in the same microenvironment [32].

Severe equine asthma, the most common lower airway inflammatory disease of mature horses, is characterized by persistent neutrophilic inflammation [35–37]. This marked airway neutrophilia suggests the presence of IL-8 [35, 38, 39], for which macrophages are one of the main sources [40]. Recently, increased surface expression of CD163 and CD206 on AMs was identified in association with exacerbation of SEA [11], suggesting that these macrophages may also have an anti-inflammatory role that like humans, is associated with the severity of

asthma [17, 18]. However, in that study tissue-resident AMs were not distinguished from infiltrated MDMs, nor was their cytokine response evaluated. Therefore, the precise involvement of AMs and MDMs in the pathogenesis of SEA remains unknown.

The current study addressed the hypothesis that equine alveolar macrophages/monocytes (AMMs) and MDMs have unique immunophenotypic and functional properties. To begin to assess the contribution of AMMs and MDMs to the response against agents known to exacerbate SEA, AMMs were obtained from BALF and MDMs were cultured from blood leukocytes of healthy horses. To discern individual responses, AMMs and MDMs were exposed to control medium and FLS, and surface marker expression and cytokine production were measured and compared.

## Materials and methods

### Horses and sampling

Six female Standardbred horses with an average age of 15.5 (± SD 4.03) years and no history of lung disease were selected from a research herd. Horses were up to date on vaccinations and anthelmintic prophylaxis, were fed grass hay and kept in an outdoor paddock with access to shelter. Physical and tracheobronchoscopic evaluations revealed no abnormalities. Baseline complete blood cell count (CBC) and serum biochemical profile results were all within reference intervals. All procedures were approved under Animal Use Protocol 3816 by the University of Guelph Animal Care Committee.

Jugular venipuncture was performed to collect consecutive 10 mL aliquots of blood into 9 EDTA tubes and 1 serum tube. For each horse, BAL was performed with standing intravenous sedation using 1.5 mL xylazine (Elanco Canada Limited) and 1 mL butorphanol tartrate (Zoetis). Approximately 450 mL of warmed sterile physiologic saline (Vetoquinol) was infused and on average 254.2 mL (± SD 40.05) of BALF was retrieved. Samples were immediately placed on ice and transported to the laboratory for processing.

### Blood sample processing

Whole blood and 1 mL of serum were submitted for a CBC and serum biochemical profile to the Animal Health Laboratory, Guelph. Peripheral mononuclear cells (PBMCs) were isolated from 70 mL of EDTA-anticoagulated blood using SepMate™ PBMC isolation tubes containing Lymphoprep™ density gradient (both from Stemcell Technologies) following the manufacturer's instructions. Separated PBMCs were resuspended in complete cell culture medium (RPMI 1640, Gibco™, Thermo Fisher Scientific, medium) supplemented with 10% heat-inactivated horse serum (Gibco™, Thermo Fisher Scientific), penicillin-streptomycin (Gibco™, Thermo-Fisher) and β-mercaptoethanol (Bio-Rad), seeded into a 75 cm$^2$ cell culture flask (Nunc™, Thermo Fisher Scientific), and incubated at 37˚C with 5% $CO_2$. After 1 hour of incubation, non-adherent lymphocytes in the supernatant were removed from the flasks, and the monocytes adhering to the flask were washed 3 times with medium. Monocytes were cultured in medium for 7 days to promote differentiation into MDMs. Medium was replaced every two days. After 7 days, the number of cells was counted (Moxi™ Z Mini Automated Cell Counter, ORFLO Technologies) and adjusted to 10$^6$ cells per well before exposure to FLS.

### Bronchoalveolar lavage fluid sample processing

The BALF samples were processed within 20 minutes of collection. Cytocentrifuge preparations were made for evaluation by light microscopy using 150 μL of BALF centrifuged for 6 minutes at 41 $g$ (Shandon-Elliott cytocentrifuge SCA-0020, Shandon Scientific Co. Ltd.),

rapidly dried and stained with a modified Wright stain (Thermo Fisher Scientific) after which a 500-cell differential count was performed. The remaining BALF was aliquoted into 50 mL conical tubes (Falcon™, Thermo Fisher Scientific) and centrifuged at 400 $g$ for 10 minutes at room temperature. Supernatant was decanted and the cell pellet was washed three times in sterile phosphate-buffered saline (PBS) and resuspended in complete medium. To purify AMMs from other alveolar leukocytes, $10^8$ cells were seeded into a 75 cm$^2$ cell culture flask (Nunc, Thermo Fisher Scientific) and cultured at 37°C with 5% $CO_2$ for 4 hours. Cell culture supernatant containing non-adherent cells was then removed and fresh complete medium was added. Adherent macrophages were then detached with TrypLE™ Select Enzyme (10X, Thermo Fisher Scientific) and counted (Moxi™ Z Mini Automated Cell Counter, ORFLO Technologies). In duplicate, wells of a 6-well cell culture plate containing 2 mL complete medium were seeded with $10^6$ AMMs. After overnight incubation, cell culture supernatant was removed, and FLS or media was added.

## Exposure material

Duplicate cultures of AMMs and MDMs were exposed to a mixture of $10^6$/mL *Aspergillus fumigatus* spores, 100 ng/mL LPS (Invitrogen™ eBioscience™, Thermo Fisher Scientific), and $10^6$/mL silica microspheres (Polysciences, Inc) [8]. This challenge material (FLS) was prepared in 2 mL serum-free RPMI. The control condition was incubation with serum-free RPMI without FLS. After 6 hours [41] of incubation at 37°C with 5% $CO_2$ [42, 43] 1 mL of cell culture supernatant was collected and stored at -80°C for cytokine quantification. Cells were enzymatically (TrypLE™ Select) detached for flow cytometric analysis.

## Flow cytometry

Detached cells were transferred into 1.5 mL microcentrifuge tubes and centrifuged at 400 $g$ for 3 minutes at 4°C. Supernatant was removed and cells were washed with 200 μL flow buffer (PBS containing 2% heat inactivated horse serum, 10 mM EDTA, and 0.2% sodium azide) at 400 $g$ for 3 minutes at 4°C. After decanting off the supernatant, samples were incubated on ice for 15 minutes with a viability dye (Zombie NIR, BioLegend). For multi-color immunostaining, each sample was sequentially incubated using previously validated antibodies [11], on ice for 15 minutes, with wash steps as described above. Antibodies included: mouse anti-human CD163 (clone Ber-Mac3, 0.005 μg/μL, Novus Biologicals) [11, 44], secondary antibody rat anti-mouse IgG1 PE-CY7 (clone M1-14D12, 0.001 μg/μL, Thermo Fisher Scientific), and mouse anti-human CD206 PE (clone 3.29B1, 1.0 μg/μL, Beckman Coulter) [11, 44, 45]. After the final wash, cell pellets were resuspended in 1 mL of flow buffer and interrogated using a FACSCanto II flow cytometer (BD) with collection of 20,000 events. Flow cytometry data were analyzed using FlowJo v.10.8.1 (BD). Initial gating ensured that only live singlets were selected for further analysis (S1 Fig). Median fluorescence intensity (MFI) was assumed to be proportional to the concentration of cell surface proteins.

## Cytokine quantification

Cytokines in supernatant samples were measured in triplicate with a Bio-Plex® 200 Multiplex Immunoassay system (Bio-Rad), using the Equine Milliplex MAP Magnetic Bead Panel (MilliporeSigma) in accordance with the manufacturer's instructions. The cytokines measured in this panel were IL-1β, IL-5, IL-8, IL-10, IL-12p70, interferon gamma (IFN-γ), and TNF-α. Cytokine data were initially processed using Belysa® analysis software (MilliporeSigma), where MFI was plotted on a five-parameter logistical regression of the standard curve to determine analyte concentration. Concentrations that were below half of MilliporeSigma's stated or

Belysaⓡ analysis software-calculated limit of detection were considered undetectable and assigned a value of zero (Fig 2).

## Statistical analysis

Analyses were performed using SAS version 9.4 (Cary, NC, USA). To account for the inter-horse variability within the experiment, a one-factor factorial in a randomized complete block design was used, with individual horses as the random block and cell treatment as the fixed-effect factor.

Residual analyses were performed to assess the ANOVA assumptions. This included testing for normality with Shapiro-Wilk, Kolmogorov-Smirnov, Cramer-von Mises, and Anderson-Darling tests, and plotting the residuals against both the predicted values and explanatory variables used in the model. These analyses were used to reveal outliers, unequal variances, or the need for data transformation. Statistical significance was set at $p \leq 0.05$.

## Results

### Bronchoalveolar lavage fluid cytology

For all horses, BALF 500-cell differential counts were interpreted to indicate absence of airway inflammation if neutrophils were $\leq 5\%$, mast cells $\leq 2\%$, and eosinophils $\leq 1\%$ [35] (S1 Table).

### Flow cytometry

For technical reasons, post-challenge MDMs from horse 1 and one sample from horse 4 were excluded from flow cytometric evaluation. Data became normally distributed after natural logarithmic transformation. In the remaining samples, and regardless of culture conditions, AMMs had significantly higher surface expression of CD163 ($p < 0.001$) and CD206 ($p < 0.001$) than MDMs (Fig 1). In AMMs, the expression of CD163 and CD206 was not altered by FLS exposure, but on MDMs, both antigens were decreased in every sample after FLS exposure, albeit in a non-statistically significant manner ($p = 0.449$ and $0.424$, respectively) (Fig 1).

### Cytokine quantification

The data for IL-8 was normal, and for all other cytokines, natural logarithmic transformation was performed. Alveolar macrophages/monocytes exposed to FLS secreted IL-1β, IFN-γ, and TNF-α while those exposed to medium alone secreted none. Although challenged AMMs produced a significantly greater amount of IL-8, control AMMs also produced IL-8. In AMMs, IL-5 and IL-10 (Fig 2, S2 Table, S2 File) were undetectable in cell culture supernatant. In MDMs, FLS stimulation resulted in increased production of IL-8, IL-10, and TNF-α compared to control MDMs. Interleukin-1β, IL-5 and INF-γ were detected neither in treated nor control MDMs (Fig 2, S2 Table, S2 File). The production of IL12p70 was below the limit of detection in all groups (S2 Table, S2 File). Between-group comparisons of cytokine production before and after FLS exposure indicated that AMMs had greater production of TNF-α, whereas MDMs had greater production of IL-8 (Fig 2, S2 Table, S2 File).

## Discussion

The current study was an in vitro comparison of the responses of healthy horse AMMs obtained from BALF and macrophages differentiated from blood monocytes to agents known to cause exacerbation of SEA in susceptible horses (FLS). In healthy horses, macrophages from BALF were 70% AMs and 30% monocytes, and those monocytes were intermediate or non-

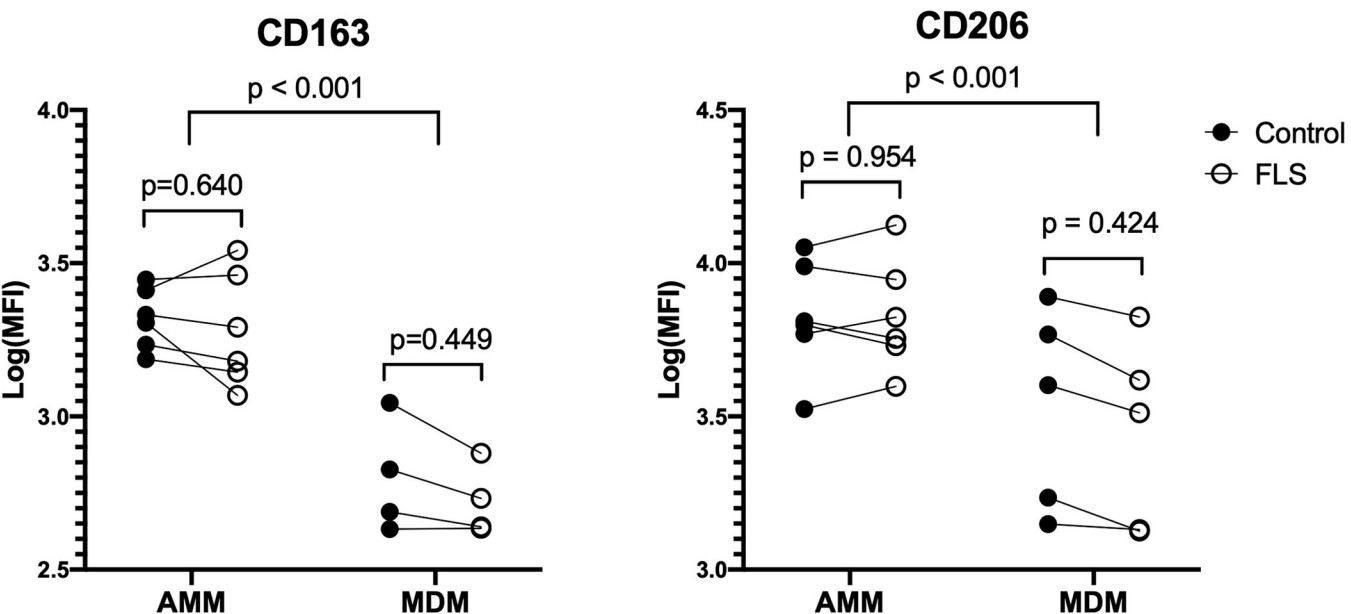

**Fig 1. Cell surface expression of CD163 and CD206 on alveolar macrophages/monocytes and monocyte-derived macrophages exposed to an inflammatory stimulus (FLS).** There was higher expression of CD163 and CD206 on AMMs than MDMs independent of culture conditions. Expression of CD163 and CD206 was not changed in FLS-exposed AMMs, and although reduced in MDMs, the decrease was not statistically significant. The median fluorescence intensity (MFI) value was log transformed.

classical [25]. In contrast, 90% of monocytes in equine whole blood were classical [23]. In addition, in our study, BALF-derived AMMs, were exposed to FLS shortly after isolation, whereas the blood-derived monocytes were cultured for 7 days to induce differentiation before exposure. Therefore, although monocytes were likely present in the BALF pool, due to their presumed low concentration, we considered BALF-derived macrophages as a bulk AM population and used it as one model.

The findings in this study confirm the hypothesis that AMMs and MDMs have differing immunophenotypes and cytokine responses to FLS exposure. By flow cytometry, significant differences in surface expression of CD163 and CD206 were identified between AMMs and MDMs as groups. With FLS stimulation, overall cytokine production in both AMMs and MDMs suggested a proinflammatory response, although cytokine patterns differed slightly between cell types. These results imply differing roles for AMMs and MDMs in initiating and modulating inflammatory responses in the early stages of lung inflammation.

In the present study, healthy horse-origin AMMs had an overall higher expression of CD163 and CD206 than MDMs. Among other functions, a major role of CD163 is to bind bacteria, and of CD206, to mediate phagocytosis of bacteria and fungi [46]. Greater expression of these receptors might indicate that AMMs are more efficient phagocytes than MDMs [47]. In other species, upregulation of both receptors indicated an anti-inflammatory phenotype [48, 49], and macrophages expressing CD163 played a role in resolution of inflammation [50]. In general, AMMs might have a greater role than MDMs in restricting inflammation, whereas MDMs may have a more proinflammatory phenotype and therefore an important role in the pathogenesis of inflammatory and degenerative lung diseases [51].

Expression of CD163 and CD206 on AMMs and MDMs was minimally affected by 6 hours' exposure to FLS. Although reduced CD163 and CD206 expression was detected on MDMs it was not statistically significant. Cell surface protein expression is affected by cytokines, and

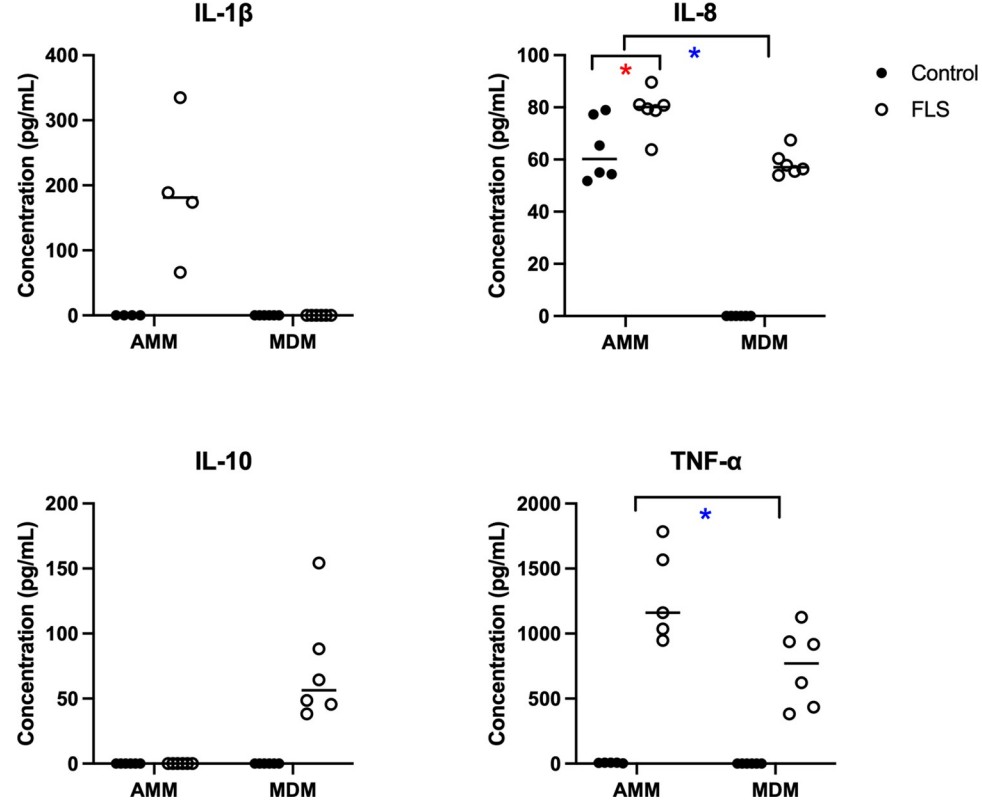

**Fig 2. Mean or median cytokine concentration in alveolar macrophage/monocyte and monocyte-derived macrophage supernatant.** Cells were incubated for 6 hours with serum-free RPMI (control) or FLS. Red asterisks indicate significant treatment effects of FLS incubation compared to control in each cell type. Blue asterisks indicate significant treatment effects of FLS between cell types. ND indicates non-detectable cytokine concentration.

changes at different stages of inflammation. For instance, human monocytes and macrophages cultured with IL-10 had upregulated CD163 expression, whereas IFN-γ exposure was associated with CD163 downregulation [14], and stimulation with LPS increased the surface expression of CD163 and decreased the expression of CD206 [13]. However, the precise mechanisms behind this regulation remain unknown [52]. In addition to the cell membrane form, CD163 and CD206 can be shed upon bacterial stimulation, generating soluble forms [53–55], which can be used as biomarkers for different diseases [56–58]. In this study, AMM and MDM responses were determined for a short interval in vitro, and the soluble forms of these proteins were not measured [11]. Therefore, the minimal effect of FLS on the expression of CD163 and CD206 might be because the short incubation time was insufficient to consistently alter the expression of cell surface proteins, or the proteins were released from the cell membrane.

Macrophages regulate the immune system by releasing cytokines. The bead-based multiplex assay used in this study allowed sensitive and simultaneous measurements of multiple cytokines in a small volume of solution [59, 60]. The individual components of the FLS challenge material may have been recognized by several receptors on macrophages, including those for signal transduction or phagocytosis. For example, activation of dectin-1 through recognition of β-glucan on fungal spores results in an increased phagocytic and proinflammatory response [27, 29]. In the current work, these effects were evidenced by secretion of multiple proinflammatory cytokines from both AMMs and MDMs. Additionally, production of the potent proinflammatory cytokine, TNF-α, by both AMMs and MDMs likely resulted from the recognition

of LPS by TLR-4, which drives signal transduction mainly via the MyD88 pathway resulting in NF-κB activation [61–63]. Finally, in humans, phagocytosis of silica by AMMs induced activation of the inflammasome, which cleaved pro-IL-1β into proinflammatory IL-1β [64]. Therefore, the silica microspheres included in the challenge material likely contributed to the secretion of IL-1β from AMMs incubated with FLS.

Macrophage behaviour is highly influenced by the presence of IFN-γ, a powerful driver of a proinflammatory macrophage phenotype [65–67], and production of IFN-γ by BALF cells was a feature of horses in exacerbation of SEA [38]. Although T cells and natural killer cells are major sources of IFN-γ, human AMs also produced IFN-γ when stimulated with IL-12 and IL-18 [68, 69]. In the present study, AMMs from four horses secreted IFN-γ in response to FLS exposure although the concentrations were below the stated limit of detection of the assay. While other researchers detected increased gene expression of IL-10 in equine AMs after hay dust challenge [42, 70], IFN-γ and IL-10 counteract one another [68, 71]. In the current study, self-supplementation with IFN-γ likely explains the absence of IL-10 in the culture supernatant from AMMs after FLS exposure. In contrast to AMMs, MDMs did not produce IFN-γ and had increased IL-10 secretion with FLS stimulation suggesting an inability to initiate IFN-γ-derived immune responses. These findings suggest that recruited monocytes may also promote an anti-inflammatory milieu in the early inflammatory response, and also highlight the importance of direct protein measurement in cell function studies.

Increased production of neutrophil chemotactic compounds such as IL-8 play an important role in the pathogenesis of SEA [38, 40, 72]. Neutrophil release of elastases and other compounds leads to the tissue damage and remodelling characteristic of SEA [73]. Because of the LPS content of FLS, it was expected that both AMMs and MDMs would have significantly increased IL-8 production after exposure. However, unstimulated AMMs had considerable innate secretion of IL-8, which increased significantly but not as markedly as post-FLS exposure MDM IL-8 production. This finding is similar to work that identified substantial basal secretion of IL-8 by human AMs that increased only moderately after stimulation with LPS [74]. Basal IL-8 secretion by AMMs differentiates them from MDMs, and might be due to the priming effect of constant inhalation of airborne agents [31]. Alternately, AMMs might be reactive to the plastic culture flask or the act of adherence itself may stimulate AMM release of IL-8 [74]; however, the effects of these latter two processes have not been studied.

The lack of IL-5 detection across cell types and exposure conditions was expected because IL-5 is not produced by macrophages [75], therefore, the inclusion of IL-5 in this cytokine panel functioned as an additional negative control.

In this study, AMMs and MDMs derived from healthy horses each displayed a distinct immunophenotype and cytokine production profile. Few studies have investigated functional and phenotypic differences between equine macrophage subpopulations. However, similar to the present work, AMs from young horses had higher TNF-α gene expression and lower IL-10 expression than MDMs after infection with virulent *Rhodococcus equi* [76]. A comparison between equine AMs and peritoneal macrophages showed that AMs had higher intrinsic surface expression of CD163 than peritoneal macrophages, and AMs had increased production of TNF-α while peritoneal macrophages did not [77]. When stimulated with LPS, AMs and blood mononuclear cells differed in their IL-8 and IL12p40 gene expression [78, 79]. These findings along with the results of the present study highlight the unique attributes of different cell types from the macrophage lineage. Therefore, future work must account for the differences between macrophage subpopulations in investigations of macrophage function in the pathogenesis of equine disease.

In conclusion, this study identified unique immunophenotypes and cytokine production in AMMs and MDMs in response to an inflammatory stimulus. Alveolar macrophages had

higher surface expression of CD163 and CD206. When stimulated with FLS, AMMs released IL-1β, IL-8, INF-γ and TNF-α, and MDMs produced IL-8, IL-10 and TNF-α, suggesting the potential for differing roles in the modulation of lower airway inflammation in horses.

## Supporting information

**S1 Fig. Flow cytometric analysis of equine alveolar macrophages after exposure to antigen challenge mixture.** Monocyte-derived macrophages were uniform based on light scatter properties, thus a flow cytometry plot with their gating strategy is not provided. A: Cells were separated from debris based on size (forward scatter, FSC) and internal complexity (side scatter, SSC). B: Singlets, i.e., cells that passed through the laser beam one at a time were selected. C: Live cells were identified based on their negativity for the viability dye. D. Unstained control. E. Dual stained sample. The challenge mixture was composed of $10^6$/mL *Aspergillus fumigatus* spores, 100 ng/mL LPS and $10^6$/mL silica microspheres in 2 mL serum-free RPMI.
(TIF)

**S1 File. Raw data for flow cytometry assay.**
(XLSX)

**S2 File. Raw data for cytokine assay.**
(XLSX)

**S1 Table. Cellular composition of bronchoalveolar lavage fluid.**
(DOCX)

**S2 Table. P-values comparing cytokine production in different cell culture conditions.**
(DOCX)

## Acknowledgments

The authors thank Dr. Dana Patcas (MilliporeSigma) for technical assistance with the cytokine assays.

## Author Contributions

**Conceptualization:** Heng Kang, Dorothee Bienzle, Brandon N. Lillie, Janet Beeler-Marfisi.

**Data curation:** Heng Kang.

**Formal analysis:** Heng Kang, William Sears.

**Funding acquisition:** Dorothee Bienzle, Janet Beeler-Marfisi.

**Investigation:** Heng Kang, Gary Kwok Cheong Lee, Luis G. Arroyo, Janet Beeler-Marfisi.

**Methodology:** Heng Kang.

**Project administration:** Heng Kang, Janet Beeler-Marfisi.

**Resources:** Janet Beeler-Marfisi.

**Supervision:** Brandon N. Lillie, Janet Beeler-Marfisi.

**Visualization:** Heng Kang.

**Writing – original draft:** Heng Kang.

**Writing – review & editing:** Gary Kwok Cheong Lee, Dorothee Bienzle, Luis G. Arroyo, Janet Beeler-Marfisi.

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
