## [Decision Letter · Decision Letter 0]

7 Oct 2022

PONE-D-22-22366Equine alveolar macrophages and monocyte-derived macrophages respond differently to an inflammatory stimulusPLOS ONE

Dear Dr. Kang,

Thank you for submitting your manuscript to PLOS ONE. After careful consideration, we feel that it has merit but does not fully meet PLOS ONE’s publication criteria as it currently stands. Therefore, we invite you to submit a revised version of the manuscript that addresses the points raised during the review process.

ACADEMIC EDITOR:  The statistical analysis was either inadequate (does not account for different horses, for example) or absent. There are many parts of the methods that were unclear. The major issue for me, however, is that the study compares 2 different cell types, but its concerning that one cell type (AMs) can be “contaminated” by the other cell type, therefore better interpretation and visualization of the data is needed.

We look forward to receiving your revised manuscript.

Kind regards,

Selvakumar Subbian, Ph.D.

Academic Editor

PLOS ONE

Journal Requirements:

Reviewers' comments:

Reviewer's Responses to Questions

**Comments to the Author**

1. Is the manuscript technically sound, and do the data support the conclusions?

Reviewer #1: No

Reviewer #2: Partly

2. Has the statistical analysis been performed appropriately and rigorously? 

Reviewer #1: No

Reviewer #2: No

3. Have the authors made all data underlying the findings in their manuscript fully available?

Reviewer #1: No

Reviewer #2: No

4. Is the manuscript presented in an intelligible fashion and written in standard English?

Reviewer #1: Yes

Reviewer #2: Yes

5. Review Comments to the Author

Reviewer #1: Comments on “Equine alveolar macrophages and monocyte-derived macrophages respond differently to an inflammatory stimulus”

Severe equine asthma is associated with exposure of the horse to A. fumigatus, spores, LPS derived from gram-negative bacteria and silica microspheres. The authors attempt to investigate the effect of these elements in vitro on alveolar macrophages and monocyte-derived macrophages. For this purpose they determine the expression levels of molecules such as CD163 and CD206 as well as a study of cytokine secretion. The question posed by the authors is very interesting and definitely of relevance and interest for the understanding of a disease that affects a susceptible population of horses.

In reviewing the document, some aspects were intriguing and I describe them below:

Line 159. The authors used TrypLE select enzyme to detach cells prior to immunostaining and flow cytometry. This reagent has a recombinant enzyme that breaks glycine and lysine bonds in proteins. The authors developed an experiment to determine if this reagent does not affect the presence of CD163 and CD206? antigens. If they performed the experiment, I would like to know about it since my concern is the lack of increased expression of these proteins following FLS stimulation as shown in Figure 2.

171 How did the authors determine that 6 hours would be the appropriate time to finish the experiment? Why did they not perform a 12, 24 and 48 hour curve? It is known that cytokine secretion minimally incubates, in the presence of stimulus, for 6 hours. However, the ideal is to determine a cytokine production curve and I believe that the same applies to the experiment associated with CD163 and CD206 markers.

Flow cytometry.

The authors make use of anti-human CD163 and anti-human CD206 PE antibodies, considering that these antibodies are designed for studies in humans and not in horses, their validation is elementary. I suggest to the authors that instead of the supplementary figure S1 they make a figure with the multiple scatter plots describing their gating strategy, that it contains the plots of cells without immunostaining, as well as the single staining controls or in its case FMO controls. The idea is to provide certainty about the intrinsic fluorescence of the cells, the basal fluorescence of the fluorophore panel minus the one to be analyzed and the effectiveness of the compensation. This would give more certainty to the results observed in Figure 2. I believe that Figure 1 does not provide more information than the gating strategy on live/dead cells.

Results.

It is recommended that Figure 1 be omitted as the same representation is found in Figure 2. Figure 1 represents basal expression levels prior to FLS challenge, therefore it can be omitted along with the associated analysis and its descriptions in the text.

Figure 2 shows that the expression levels of the CD163 marker is not affected after FLS challenge in both AMs and MDMs. On the other hand, the CD206 marker, in the case of AMs is not affected while in MDMs it is slightly reduced and although statistically significant, very close to non-significant after FLS challenge. Could it be possible that the use of the TriPLE reagent impacted the surface antigen conformation of both CD markers? This coupled with the fact that the antibodies are designed to recognize human CDs, is it possible that the affinity of the antibodies for recognition of these CDs could be lost? Could this slight decrease be considered to be due to the small number of samples tested? An n=5 may not be representative. It is suggested to the authors to mention if they used standard deviation or standard error of the mean. On the other hand, it is suggested to use a dot plot.

Line 246, it is suggested to the authors to remove the texts where some cytokine is mentioned but it is not statistically significant and only mention its non-significance or non-presence in a final line.

Line

Discussion.

Considering the data presented in Figure 1 it is difficult to conclude that there are significant differences in the expression levels of the analyzed CDs. Except for the slight reduction in CD206 in MDM cells after stimulation, this may be due to a very low sample population or a problem with the cell detachment technique or overcompensation during analysis in Flow Jo. The results are inconsistent since it has been reported that CD163 protein is downregulated in the presence of inflammatory processes but during the cytokine experiment IL-10 is observed to be elevated. How this is explained, in turn the findings are incongruent with respect to the findings found in reference 11, which is a publication of the same authors.

An additional explanation as to why the authors do not identify significant differences with respect to the CD163 marker is that CD163 is secreted to the medium in a soluble form when stimulated. Why did the authors not quantify this form of the protein? The same case occurs with the CD206 marker, which in other studies is analyzed in its soluble form instead of its membrane form.

(Tsuchiya, K., Suzuki, Y., Yoshimura, K. et al. Macrophage Mannose Receptor CD206 Predicts Prognosis in Community-acquired Pneumonia. Sci Rep 9, 18750 (2019). https://doi.org/10.1038/s41598-019-55289-2).

Biomarkers in Acute Kidney Injury

C.L. Edelstein MD, PhD, in Biomarkers of Kidney Disease (Second Edition), 2017

Cerebrospinal Fluid in Neurologic Disorders

Clara Matute-Blanch, ... Manuel Comabella, in Handbook of Clinical Neurology, 2018

Immunotherapy of Cancer

Xiaofei Yu, ... Xiang-Yang Wang, in Advances in Cancer Research, 2015

Line 274. "greater expression... "lacks citation.

Line 352-358. Can a basal secretion state be considered a predisposing factor to an inflammatory process? The constitutive expression of IL-12p70 is not altered in the presence of the stimulus, therefore I consider that it is not relevant and suggestive of an inflammatory state.

It is suggested to the authors to shorten the amount of explanatory information in the justification section and to perform a more critical analysis based on the actual findings identified in the study and considering a more rigid analysis of the data.

Reviewer #2: Dear editor and authors,

I read with interest the present paper on the different responses of AMs and MDMs to FLS. Please find below a few comments to consider.

To account for inter-individual variation, the figures need to include data from all horses before and after in vitro exposure (rather than mean), similar to Figure 6 in the authors previous publication (Kang et al., Flow cytometric analysis of equine bronchoalveolar lavage fluid cells in horses with and without severe equine asthma, Vet Path, 2021).

Why were those specific cytokines selected? For example, why not IL-4?

Line 85-86: “under homeostatic conditions, the majority of macrophages in alveoli are AMs and the minority are MDMs.” In this study, the differences of AM and MDMs are explored, but it was assumed all macrophages in BALF were AM. In Sage et al. (https://www.frontiersin.org/articles/10.3389/fimmu.2022.929922/full) the authors describe the presence of monocytes in BALF, please include this paper in your discussion.

There are additional important references as far as differences between AMs and MDMs that were overlooked (for ex: Berghaus et al 2014 https://pubmed.ncbi.nlm.nih.gov/24736188/). Please perform a thorough Pubmed search, and include pertinent references that may not be related to SEA, but to horses.

Please provide more details on the methods:

o Was 10% horse serum heat-inactivated or non-heat-inactivated?

o Was the number of cells adjusted after incubation and before FLS challenge? For example, was there an extra well for cell counting?

o How was the purity of the monocyte fraction determined?

o How were the cells detached for flow cytometric analysis?

o Data analysis

Flow cytometry: details of statical analysis missing

How was the inter-individual variation included in the analysis? In other words, how were the differences among horses/experiments/days account for?

Was there statistical analysis performed on the number of AMs? (differences among horses, AMs were 70% for some, 50% for others)

Are data from Fig. 1 the same as Fig. 2?

I do not know where the antibody would bind on CD206, but could it be that the lower expression of CD206 on MDMs after the exposure to FLS be because of binding sites being blocked by the binding of FLS?

6. PLOS authors have the option to publish the peer review history of their article (what does this mean?). If published, this will include your full peer review and any attached files.

Reviewer #1: No

Reviewer #2: No

---

## [Author Response · Author response to Decision Letter 0]

19 Jan 2023

ACADEMIC EDITOR: The statistical analysis was either inadequate (does not account for different horses, for example) or absent. There are many parts of the methods that were unclear. The major issue for me, however, is that the study compares 2 different cell types, but its concerning that one cell type (AMs) can be “contaminated” by the other cell type, therefore better interpretation and visualization of the data is needed.

Thank you for pointing out these issues. Based on the Reviewers’ comments, a new statistical analysis was performed. To account for inter-horse differences, a one-factor factorial in a randomized compete block design was used. Please see the updated statistical analysis section for a detailed description.

We acknowledge the presence of monocytes in the BALF-derived macrophage pool. We have added a description of this issue in the discussion. 

Journal Requirements:

We have reviewed the requirements, and modified the file names accordingly.

We have included all the original data in the resubmission. Therefore, “data not shown” was replaced with “S2 File”.

Reviewers' comments:

Reviewer's Responses to Questions

Comments to the Author

1. Is the manuscript technically sound, and do the data support the conclusions?

Reviewer #1: No

Reviewer #2: Partly

Addressed, as noted above.

2. Has the statistical analysis been performed appropriately and rigorously?

Reviewer #1: No

Reviewer #2: No

Addressed, as noted above.

3. Have the authors made all data underlying the findings in their manuscript fully available?

Reviewer #1: No

Reviewer #2: No

Addressed, as noted above.

4. Is the manuscript presented in an intelligible fashion and written in standard English?

Reviewer #1: Yes

Reviewer #2: Yes

5. Review Comments to the Author

Reviewer #1: Comments on “Equine alveolar macrophages and monocyte-derived macrophages respond differently to an inflammatory stimulus”

Severe equine asthma is associated with exposure of the horse to A. fumigatus, spores, LPS derived from gram-negative bacteria and silica microspheres. The authors attempt to investigate the effect of these elements in vitro on alveolar macrophages and monocyte-derived macrophages. For this purpose they determine the expression levels of molecules such as CD163 and CD206 as well as a study of cytokine secretion. The question posed by the authors is very interesting and definitely of relevance and interest for the understanding of a disease that affects a susceptible population of horses.

In reviewing the document, some aspects were intriguing and I describe them below:

Line 159. The authors used TrypLE select enzyme to detach cells prior to immunostaining and flow cytometry. This reagent has a recombinant enzyme that breaks glycine and lysine bonds in proteins. The authors developed an experiment to determine if this reagent does not affect the presence of CD163 and CD206? antigens. If they performed the experiment, I would like to know about it since my concern is the lack of increased expression of these proteins following FLS stimulation as shown in Figure 2.

We did not develop an experiment to test the effects of TrypLE select enzyme on the surface expression of CD163 and CD206. However, TrypLE select enzyme preserved cell-surface epitope expression including CD2 (ThermoFisher) and CD24 (Panchision et al. 2007). In addition, both the control groups and FLS-stimulated groups were treated with TrypLE select enzyme – FLS was the only different condition between groups. 

https://static.fishersci.eu/content/dam/fishersci/en_US/documents/programs/scientific/brochures-and-catalogs/brochures/unrivaled-live-sciences-essentials-q4-brochure-17-049-2027.pdf

Panchision DM, Chen HL, Pistollato F, Papini D, Ni HT, Hawley TS. Optimized flow cytometric analysis of central nervous system tissue reveals novel functional relationships among cells expressing CD133, CD15, and CD24. Stem Cells. 2007;25(6):1560-1570. doi:10.1634/stemcells.2006-0260

171 How did the authors determine that 6 hours would be the appropriate time to finish the experiment? Why did they not perform a 12, 24 and 48 hour curve? It is known that cytokine secretion minimally incubates, in the presence of stimulus, for 6 hours. However, the ideal is to determine a cytokine production curve and I believe that the same applies to the experiment associated with CD163 and CD206 markers.

The 6 hour incubation time was determined based on gene expression curves previously determined in our lab (Odemuyiwa, 2012). This citation was added to the manuscript:

 Odemuyiwa, SO. Immunophenotypic Characteristics of Equine Monocytes and Alevolar Macrophages. Ph.D. Dissertation. University of Guelph. 2012. Available at https://atrium.lib.uoguelph.ca/xmlui/handle/10214/3631

Also, we used live fungal spores in our challenge material. The spores germinated around 10 hours of incubation; therefore, an incubation time longer than 10 hours was not possible in this experiment. Common methods for inactivating fungal spores cause macrophages to respond to them differently (Hohl et al, 2005 10.1371/journal.ppat.0010030). In our hands, commercially available sources of β-glucan do not stimulate macrophages the same way that viable spores do. We are currently assessing our own β-glucan lysate with the goal of replacing viable spores in future studies.

However, we believe that in an in vitro monoculture environment, a longer incubation time provides less meaningful biological information because of the lack of participation of other immune cells and lung tissue. 

Flow cytometry.

The authors make use of anti-human CD163 and anti-human CD206 PE antibodies, considering that these antibodies are designed for studies in humans and not in horses, their validation is elementary. I suggest to the authors that instead of the supplementary figure S1 they make a figure with the multiple scatter plots describing their gating strategy, that it contains the plots of cells without immunostaining, as well as the single staining controls or in its case FMO controls. The idea is to provide certainty about the intrinsic fluorescence of the cells, the basal fluorescence of the fluorophore panel minus the one to be analyzed and the effectiveness of the compensation. This would give more certainty to the results observed in Figure 2. I believe that Figure 1 does not provide more information than the gating strategy on live/dead cells.

1. The two specific clones of antibodies were previously validated in horse studies. Kang et al, 2022, in specific, addresses the above points. The following references were added to the manuscript.

Kang H, Bienzle D, Lee GKC, et al. Flow cytometric analysis of equine bronchoalveolar lavage fluid cells in horses with and without severe equine asthma. Vet Pathol. 2022;59(1):91-99. 

Steinbach F, Stark R, Ibrahim S, et al. Molecular cloning and characterization of markers and cytokines for equid myeloid cells. Vet Immunol Immunopathol. 2005;108(1-2):227-236.

Ziegler A, Everett H, Hamza E, et al. Equine dendritic cells generated with horse serum have enhanced functionality in comparison to dendritic cells generated with fetal bovine serum. BMC Vet Res. 2016;12(1):254. Published 2016 Nov 15. 

2. We agree with the suggestions on S1 Figure. It has been updated accordingly.

Results.

It is recommended that Figure 1 be omitted as the same representation is found in Figure 2. Figure 1 represents basal expression levels prior to FLS challenge, therefore it can be omitted along with the associated analysis and its descriptions in the text.

Figure 1 has been omitted as suggested. 

Figure 2 shows that the expression levels of the CD163 marker is not affected after FLS challenge in both AMs and MDMs. On the other hand, the CD206 marker, in the case of AMs is not affected while in MDMs it is slightly reduced and although statistically significant, very close to non-significant after FLS challenge. Could it be possible that the use of the TriPLE reagent impacted the surface antigen conformation of both CD markers? This coupled with the fact that the antibodies are designed to recognize human CDs, is it possible that the affinity of the antibodies for recognition of these CDs could be lost? Could this slight decrease be considered to be due to the small number of samples tested? An n=5 may not be representative. It is suggested to the authors to mention if they used standard deviation or standard error of the mean. On the other hand, it is suggested to use a dot plot.

After the new statistical analysis, the decreased expression of CD206 on MDMs was non-significant same as the reviewer observed. It is unlikely that the affinity of these antibodies was lost because compared to the unstained control, these antibodies showed positivity on the cells (S1 Figure). The small sample size might also be a reason for the non-significant results, this is one of the limitations of the study, which has been addressed in the discussion.

The figure was reformatted using dot plots to represent each horse as suggested. (Figure 1)

Line 246, it is suggested to the authors to remove the texts where some cytokine is mentioned but it is not statistically significant and only mention its non-significance or non-presence in a final line.

These points have been removed as suggested. The paragraph has been edited. 

Line

Discussion.

Considering the data presented in Figure 1 it is difficult to conclude that there are significant differences in the expression levels of the analyzed CDs. Except for the slight reduction in CD206 in MDM cells after stimulation, this may be due to a very low sample population or a problem with the cell detachment technique or overcompensation during analysis in Flow Jo. The results are inconsistent since it has been reported that CD163 protein is downregulated in the presence of inflammatory processes but during the cytokine experiment IL-10 is observed to be elevated. How this is explained, in turn the findings are incongruent with respect to the findings found in reference 11, which is a publication of the same authors.

An additional explanation as to why the authors do not identify significant differences with respect to the CD163 marker is that CD163 is secreted to the medium in a soluble form when stimulated. Why did the authors not quantify this form of the protein? The same case occurs with the CD206 marker, which in other studies is analyzed in its soluble form instead of its membrane form.

(Tsuchiya, K., Suzuki, Y., Yoshimura, K. et al. Macrophage Mannose Receptor CD206 Predicts Prognosis in Community-acquired Pneumonia. Sci Rep 9, 18750 (2019). https://doi.org/10.1038/s41598-019-55289-2).

Biomarkers in Acute Kidney Injury

C.L. Edelstein MD, PhD, in Biomarkers of Kidney Disease (Second Edition), 2017

Cerebrospinal Fluid in Neurologic Disorders

Clara Matute-Blanch, ... Manuel Comabella, in Handbook of Clinical Neurology, 2018

Immunotherapy of Cancer

Xiaofei Yu, ... Xiang-Yang Wang, in Advances in Cancer Research, 2015

The results from the flow cytometry experiment were not what we expect. There might be few reasons for that, such as small sample size, detachment technique, or a short incubation time. The inconsistent findings across different experiments might be that the in vitro cell culture condition did not represent the in vivo immune networks, and a shorter exposure time. 

We only measured the surface expression of CD163 and CD206 because the membrane form of these proteins is considered an anti-inflammatory macrophage marker. However, in the discussion, we should have not omitted the potential of shedding of these proteins. This point has been added.

Thank you for the references. We have integrated this information in the manuscript for a more comprehensive discussion. 

Line 274. "greater expression... "lacks citation.

Thank you. A citation was added.

Line 352-358. Can a basal secretion state be considered a predisposing factor to an inflammatory process? The constitutive expression of IL-12p70 is not altered in the presence of the stimulus, therefore I consider that it is not relevant and suggestive of an inflammatory state.

We agree with the reviewer’s opinion, which unchanged secretion should not indicate inflammation. This paragraph was removed.

It is suggested to the authors to shorten the amount of explanatory information in the justification section and to perform a more critical analysis based on the actual findings identified in the study and considering a more rigid analysis of the data.

We have shortened the explanatory information and added more discussions of our findings in comparison with other work. A new statistical analysis was developed. Please see the updated statistical analysis section for a detailed description.

Reviewer #2: Dear editor and authors,

I read with interest the present paper on the different responses of AMs and MDMs to FLS. Please find below a few comments to consider.

To account for inter-individual variation, the figures need to include data from all horses before and after in vitro exposure (rather than mean), similar to Figure 6 in the authors previous publication (Kang et al., Flow cytometric analysis of equine bronchoalveolar lavage fluid cells in horses with and without severe equine asthma, Vet Path, 2021).

Thank you for the suggestion. The flow cytometry figure was reformatted using dot plots to represent each horse. However, the cytokine figure cannot be presented in this way because many samples were “non-detects” therefore share the value 0. Instead, the cytokine figure was reformatted using mean/median and the confidence interval. 

Why were those specific cytokines selected? For example, why not IL-4?

The cytokines were selected based on our interest in the production of pro- or anti-inflammatory cytokines by macrophages and our budget as the full panel was more than we could afford at the time. We agree that the full panel of cytokines would have resulted in a more comprehensive better picture of macrophage response. Including the full panel is one of our future directions. 

Line 85-86: “under homeostatic conditions, the majority of macrophages in alveoli are AMs and the minority are MDMs.” In this study, the differences of AM and MDMs are explored, but it was assumed all macrophages in BALF were AM. In Sage et al. (https://www.frontiersin.org/articles/10.3389/fimmu.2022.929922/full) the authors describe the presence of monocytes in BALF, please include this paper in your discussion.

Thank you for the recommending this paper. This limitation has now been considered and added to the discussion. 

There are additional important references as far as differences between AMs and MDMs that were overlooked (for ex: Berghaus et al 2014 https://pubmed.ncbi.nlm.nih.gov/24736188/). Please perform a thorough Pubmed search, and include pertinent references that may not be related to SEA, but to horses.

Thank you for recommending this important paper. We have expanded our discussion to include all relevant equine macrophage papers. 

Please provide more details on the methods:

o Was 10% horse serum heat-inactivated or non-heat-inactivated?

It was heat-inactivated, this information has been added.

o Was the number of cells adjusted after incubation and before FLS challenge? For example, was there an extra well for cell counting?

The number of cells was counted and adjusted to one million cells per well before the FLS challenge. This information was included in the AM section but not the MDM section from the original manuscript. It has now been added to the MDM section.

o How was the purity of the monocyte fraction determined?

The cells attached to the cell culture dishes were detached and cytocentrifuged. The purity of monocytes was determined by the light microscopic evaluation, and was 100% in all cases. 

o How were the cells detached for flow cytometric analysis?

The cells were enzymatically (TrypLE™ Select) detached for flow cytometric analysis. The reagent information was added to this section. 

o Data analysis

Flow cytometry: details of statical analysis missing

How was the inter-individual variation included in the analysis? In other words, how were the differences among horses/experiments/days account for?

Thank you for pointing out these statistical issues. As noted above, to account for inter-horse variability, a one-factor factorial in a randomized compete block design was used. Please see the updated statistical analysis section for the detailed description.

Was there statistical analysis performed on the number of AMs? (differences among horses, AMs were 70% for some, 50% for others)

We did not perform statistical analyses for the cellular percentages between horses because it was not part of the scheme of the study. We were more interested in cellular responses to challenge material in vitro. We agree that cell sources were also important, therefore, we performed a complete respiratory exam and BAL cytological evaluation to ensure all horses were healthy and cytologically “quiet”. 

Are data from Fig. 1 the same as Fig. 2?

Yes, they are. The figures were reformatted in accordance with both Reviewers’ suggestions. 

I do not know where the antibody would bind on CD206, but could it be that the lower expression of CD206 on MDMs after the exposure to FLS be because of binding sites being blocked by the binding of FLS?

The surface CD206 protein contains a cysteine-rich domain, a fibronectin type II domain, and 8 eight C-type lectin domains. These domains do not bind the material from FLS. Although it was reported that CD206 could recognize LPS from Klebsiella pneumoniae (Zamze, 2002), no studies have found it could bind soluble LPS. After a thorough review, we did not find out the specific domain the anti-CD206 antibody binds. However, blocking did not seem to be occurring because the expression of CD206 on AMs did not decrease after FLS exposure.

---

## [Decision Letter · Decision Letter 1]

9 Feb 2023

PONE-D-22-22366R1Equine alveolar macrophages and monocyte-derived macrophages respond differently to an inflammatory stimulusPLOS ONE

Dear Dr. Kang,

Thank you for submitting your manuscript to PLOS ONE. After careful consideration, we feel that it has merit but does not fully meet PLOS ONE’s publication criteria as it currently stands. Therefore, we invite you to submit a revised version of the manuscript that addresses the points raised during the review process.

ACADEMIC EDITOR: Although the authors have addressed most of the comments from the reviewers, there is one issue that needs to be fixed. I agree with the comments of reviewer-2 that the cytokine data that were "ND" or below the detection limit poses challenges in statistical calculations as presented in figure-2.  Therefore, i suggest the authors to plot individual data points (of those samples that were at/above the limit of detection) in Figure-2.

We look forward to receiving your revised manuscript.

Kind regards,

Selvakumar Subbian, Ph.D.

Academic Editor

PLOS ONE

Journal Requirements:

Reviewers' comments:

Reviewer's Responses to Questions

**Comments to the Author**

1. If the authors have adequately addressed your comments raised in a previous round of review and you feel that this manuscript is now acceptable for publication, you may indicate that here to bypass the “Comments to the Author” section, enter your conflict of interest statement in the “Confidential to Editor” section, and submit your "Accept" recommendation.

Reviewer #2: (No Response)

2. Is the manuscript technically sound, and do the data support the conclusions?

Reviewer #2: Partly

3. Has the statistical analysis been performed appropriately and rigorously? 

Reviewer #2: No

4. Have the authors made all data underlying the findings in their manuscript fully available?

Reviewer #2: Yes

5. Is the manuscript presented in an intelligible fashion and written in standard English?

Reviewer #2: Yes

6. Review Comments to the Author

Reviewer #2: The authors addressed most of my concerns, with exception of the 2 below (1 new, minor).

1) The authors use throughout the paper the term cytokine multiplex measurement. Multiplex measurement just means multiple cytokines detected simultaneously, not the actual assay. Please consider using the term "bead-based multiplex assay" rather than multiplex measurement.

2) Authors: The flow cytometry figure was reformatted using dot plots to represent each horse. However, the cytokine figure cannot be presented in this way because many samples were “non-detects” therefore share the value 0. Instead, the cytokine figure was reformatted using mean/median and the confidence interval.

Thank you for the work on flow cytometry figures.

In the cytokines, if there were many “non-detects”, I believe the use of mean/median and confidence interval is not correct. In fact, in the Millipore Sigma website (link below) of the Milliplex MAP assay, one will find all sensitivities for the cytokines tested. When comparing to the raw data presented (thank you for sharing them), many values that are included in the analysis are actually below the sensitivity of the assay (essentially, also “non-detected” or “0”). In my opinion, the authors do not need statistics to demonstrate that there is stimulation of cytokines if they go from being undetected detected to a certain number (leave that to the Editor). For example, Figure 2, IL-1b. The limit of detection is 29.3 pg/ml. In the table with the raw data, if I am looking correctly, all controls except for 1 (30.83 pg/ml) are below the limit of detection (one could argue that 30.83 is too close to be considered valid). Similarly, in the treated group, if the authors remove the values below limit of detection (3.68 and 24.11), the median of the treated group will be higher. Please do this exercise for all cytokines, and add the individual values similar to Figure 1 (which now looks beautiful). I understand the frustration of having to report “non-detected” for many cytokines (for example, IL-12, INFg, and most of IL-8 data are below sensitivity). That is, however, part of reporting science and it may guide future research. An alternative approach is to concentrate the samples and try to measure again, but I understand that it may not be possible.

https://www.emdmillipore.com/US/en/product/MILLIPLEX-MAP-Equine-Cytokine-Chemokine-Magnetic-Bead-Panel-Immunology-Multiplex-Assay,MM_NF-EQCYTMAG-93K

7. PLOS authors have the option to publish the peer review history of their article (what does this mean?). If published, this will include your full peer review and any attached files.

Reviewer #2: No

---

## [Author Response · Author response to Decision Letter 1]

20 Feb 2023

ACADEMIC EDITOR: Although the authors have addressed most of the comments from the reviewers, there is one issue that needs to be fixed. I agree with the comments of reviewer-2 that the cytokine data that were "ND" or below the detection limit poses challenges in statistical calculations as presented in figure-2. Therefore, I suggest the authors to plot individual data points (of those samples that were at/above the limit of detection) in Figure-2.

We have changed Figure 2 which now shows individual data points. Please see our responses to Reviewer 2 below.

Journal Requirements:

No changes to the reference list were made because no retracted papers have been cited. The search was performed using this database: http://retractiondatabase.org/RetractionSearch.aspx#?auth%3dCavarra. 

Comments to the Author

1. If the authors have adequately addressed your comments raised in a previous round of review and you feel that this manuscript is now acceptable for publication, you may indicate that here to bypass the “Comments to the Author” section, enter your conflict of interest statement in the “Confidential to Editor” section, and submit your "Accept" recommendation.

Reviewer #2: (No Response)

2. Is the manuscript technically sound, and do the data support the conclusions?

Reviewer #2: Partly

3. Has the statistical analysis been performed appropriately and rigorously?

Reviewer #2: No

4. Have the authors made all data underlying the findings in their manuscript fully available?

Reviewer #2: Yes

5. Is the manuscript presented in an intelligible fashion and written in standard English?

Reviewer #2: Yes

6. Review Comments to the Author

Reviewer #2: The authors addressed most of my concerns, with exception of the 2 below (1 new, minor).

1) The authors use throughout the paper the term cytokine multiplex measurement. Multiplex measurement just means multiple cytokines detected simultaneously, not the actual assay. Please consider using the term "bead-based multiplex assay" rather than multiplex measurement.

Thank you for the suggestion. This term was changed accordingly throughout the manuscript. 

2) Authors: The flow cytometry figure was reformatted using dot plots to represent each horse. However, the cytokine figure cannot be presented in this way because many samples were “non-detects” therefore share the value 0. Instead, the cytokine figure was reformatted using mean/median and the confidence interval.

Thank you for the work on flow cytometry figures.

In the cytokines, if there were many “non-detects”, I believe the use of mean/median and confidence interval is not correct. In fact, in the Millipore Sigma website (link below) of the Milliplex MAP assay, one will find all sensitivities for the cytokines tested. When comparing to the raw data presented (thank you for sharing them), many values that are included in the analysis are actually below the sensitivity of the assay (essentially, also “non-detected” or “0”). In my opinion, the authors do not need statistics to demonstrate that there is stimulation of cytokines if they go from being undetected detected to a certain number (leave that to the Editor). For example, Figure 2, IL-1b. The limit of detection is 29.3 pg/ml. In the table with the raw data, if I am looking correctly, all controls except for 1 (30.83 pg/ml) are below the limit of detection (one could argue that 30.83 is too close to be considered valid). Similarly, in the treated group, if the authors remove the values below limit of detection (3.68 and 24.11), the median of the treated group will be higher. Please do this exercise for all cytokines, and add the individual values similar to Figure 1 (which now looks beautiful). I understand the frustration of having to report “non-detected” for many cytokines (for example, IL-12, INFg, and most of IL-8 data are below sensitivity). That is, however, part of reporting science and it may guide future research. An alternative approach is to concentrate the samples and try to measure again, but I understand that it may not be possible.

Thank you for the thoughtful comments, also for pointing out the sensitivity information with the link. We have changed Figure 2, which now shows individual values of each sample.

We agree that many of the concentrations were below the sensitivity of the assay and have removed some cytokines, for example IL-12 and IFN-g. However, we also note that the website gives a high cut-off point for each cytokine, which might lead us to omit important information. In addition, the limit of detection (LoD) given by the website is different from the LoD calculated based on the standard curve created by the Belysa® Immunoassay Curve Fitting Software (see link below). For IFNg, the standard curve gave an LoD of 14.96 pg/mL (cell F165 from the raw data file) whereas the website indicates the Lod of IFNg is 165 pg/mL. Therefore, we decided to keep some values which are just below the sensitivity concentration but still appear meaningful. For example, the LoD of IL-8 is 74.3 pg/mL according to the website, so we kept those samples with concentrations of IL-8 around 60 pg/mL, and removed those with extremely low values of ~ 10 pg/mL. We performed this exercise for all cytokines, and hope this is a rational justification that adequately addresses your concerns.

---

## [Editor Report · Decision Letter 2]

22 Feb 2023

Equine alveolar macrophages and monocyte-derived macrophages respond differently to an inflammatory stimulus

PONE-D-22-22366R2

Dear Dr. Kang,

We’re pleased to inform you that your manuscript has been judged scientifically suitable for publication and will be formally accepted for publication once it meets all outstanding technical requirements.

Kind regards,

Selvakumar Subbian, Ph.D.

Academic Editor

PLOS ONE
---

## [Editor Report · Acceptance letter]

6 Mar 2023

PONE-D-22-22366R2 

Equine alveolar macrophages and monocyte-derived macrophages respond differently to an inflammatory stimulus 

Dear Dr. Kang:

I'm pleased to inform you that your manuscript has been deemed suitable for publication in PLOS ONE. Congratulations! Your manuscript is now with our production department. 

Kind regards, 

on behalf of

Dr. Selvakumar Subbian 

Academic Editor

PLOS ONE